# Highly Stable Hybrid Pigments Prepared from Organic Chromophores and Fluorinated Hydrotalcites

Magali Hernández [1,*], Carlos Felipe [2], Ariel Guzmán-Vargas [3,*], José Luis Rivera [4] and Enrique Lima [5,*]

1 Departamento de Ingeniería y Tecnología, Facultad de Estudios Superiores Cuautitlán, Universidad Nacional Autónoma de México, Av. 1 de Mayo s/n, Cuautitlán Izcalli 54740, Edo. de Méx., Mexico

2 Departamento de Biociencias e Ingeniería, CIIEMAD-IPN, Ciudad de México 07340, CDMX, Mexico; cfelipe@ipn.mx

3 Instituto Politécnico Nacional—ESIQIE, Avenida IPN UPALM Edificio 7, Zacatenco, Mexico City 07738, DF, Mexico

4 Facultad de Ciencias Físico-Matemáticas, Universidad Michoacana de San Nicolás de Hidalgo, Morelia 58000, Mor., Mexico; jlrivera@umich.mx

5 Laboratorio de Fisicoquímica y Reactividad de Superficies (LaFReS), Instituto de Investigaciones en Materiales, Universidad Nacional Autónoma de México, Circuito exterior s/n, Cd. Universitaria, Del. Coyoacán, Ciudad de México 04510, CP, Mexico

* Correspondence: magic140288@hotmail.com (M.H.); aguzmanv@ipn.mx (A.G.-V.); lima@iim.unam.mx (E.L.); Tel.: +52-5556224640 (E.L.)

**Abstract:** Structural hydroxide groups in layered magnesium–aluminum double hydroxides were partially replaced by fluoride ions. Fluorinated and fluorine-free materials were used as hosts for two dyes, carminic acid and hydroxyl naphthol blue, resulting in a hybrid pigment color palette. The pigments were produced by two ways, either incorporating chromophore during the synthesis of the layered double hydroxide or in a post-synthesis step through the memory effect of the LDHs. Additionally, the pigments were protected with a magnesium hydroxide phase to prevent the color from fading over time. The pigments were stable for periods as long as 10 years. The color properties of the pigments were significantly influenced by the host of dye since the presence of fluorine directly influences the acid–base properties of the layered double hydroxides. The pigments conferred their color to white cream in the preparation of colored creams. The colored creams acquired the color of the layered pigment.

**Keywords:** hydrotalcite; layered double hydroxides; pigment; color; dyes; hybrid materials





## 1. Introduction

The color industry creates dyes and pigments that, in turn, are the raw materials for other manufacturing fields, such as textiles, food, cosmetics, and paint companies. The demand for dyes and pigments among these huge industries, as well as artisans and artists, stimulates the constant development of new colorful materials [1–3]. Materials acting as pigments are solid materials providing color to the object on which they are incorporated. Pigments are compatible with the substrate they are used on but insoluble with the medium. Depending on the final application of the pigment, the requirements are more or less severe. For example, if they are proposed to provide color to food or cosmetics, their toxicity, thermal stability, and dispersion in a specific medium [4–6] will be crucial characteristics to keep in mind. Actually, the most used pigments in the cosmetics industry are organic, and they are used in the form of lakes and toners [7]. However, sometimes, pure organic or inorganic pigments are insufficient for achieving the ultimate aim. Fortunately, hybrid materials can be designed from various components in order to achieve original properties in a single material [8–10]. In this context, it is documented that hybrid pigments can be prepared from several sources of color, such as anthocyanins, betalains, and azoic dyes, among others [11,12]. These dyes are stabilized onto the surface of an inorganic

matrix, and then pigments with unique properties emerge [13,14]. Earlier, it was pointed out that layered double hydroxides act as hosts of azoic dyes to produce stable hybrid pigments [15]. After that, several works confirmed the versatility of LDHs to produce original pigments [16–19]. Some additional works proposed other inorganic matrices and also micelle media in order to stabilize both natural and synthetic chromophores [20]. Emphasizing LDHs as hosts to stabilize dyes, the matrix is particularly interesting as its physical–chemical properties can be easily modulated. Carminic acid and hydroxinaphtol blue are two dyes that find applications in the formulation of inks for ink jet printing because they can provide brilliant images. These dyes show a good solubility in essentially aqueous ink liquid. Carminic acid, particularly, has been widely used to color foods, textiles, beverages, pharmaceuticals, and cosmetics products. Both dyes are soluble in liquid media. However, they have not found applications as pigments.

Thus, we started this work with the goal to find evidence that modification of LDH, and even if this modification is minor, changes significantly its properties to host chromophores carminic acid and hydroxinaphtol blue molecules. We show at end the use of pigment hybrid based on LDH as the source of color of stable creams that can be potentially used in the cosmetic or food industry.

## 2. Materials and Methods

### 2.1. Materials

Carbonate-containing Mg–Al LDHs with a Mg–Al atomic ratio close to 3 were synthesized by a sol–gel technique from magnesium ethoxide and aluminum tri-sec-butoxide. Aluminum tri-sec-butoxide (ATB, Aldrich, St. Louis, MO, USA, 99.9%) dissolved in ethanol was refluxed and stirred for 1 h. Subsequently, the temperature was lowered to 0 °C, and $HNO_3$ was added. The mixture was stirred for 1 h. After that, magnesium methoxide (Aldrich, 99%) dissolved in butanol and water were dropped into the solution until a gel was formed, which was dried at 70 °C. The molar amounts of the reagents and solvents were as follows: EtOH = $1.54 \times 10^{-1}$, ATB = $2.7 \times 10^{-3}$, $HNO_3$ = $7.725 \times 10^{-5}$, MetMg = $5.76 \times 10^{-2}$.

Fluorinated LDH was obtained when, during the synthesis [21], a part of ATB was replaced by sodium hexa-fluoroaluminate, $Na_3AlF_6$ ($1.018 \times 10^{-3}$ molar). The nominal ratio of Mg–Al was always equal to 3. The full characterization of the fluorinated LDH has been published previously [21].

Both LDH and fluorinated LDH were used as hosts of two chromophores (carminic acid or hydroxy naphthol blue) in order to produce hybrid pigments. The chromophore was incorporated into LDH either during the LDH synthesis or using the memory effect of LDH. In the first case, during the synthesis, just before the formation of gel a solution, $2.11 \times 10^{-3}$ M of carminic aid or hydroxyl naphthol blue was dropped. A modification was also tested: after 5 h, added the chromophore magnesium methoxide was dropped in order to have a pigment covered with $Mg(OH)_2$.

For the production of pigments through the memory effect, LDH was subjected to thermal treatment at 400 °C, and then the layered structure was recovered by placing the thermally treated samples into contact with a chromophore solution.

All the colored samples were rinsed with distilled water. The amount of dye in the washes was quantified by UV-vis spectroscopy in order to estimate the amount of dye contained in the pigment. Table 1 summarizes the pigments under study. All the selected pigments in this work contained close to 4% wt of dye (CA or HNB). Other pigments with higher or lower amounts of dye were discarded in this study.

**Table 1.** Summary of pigments, including their components, prepared in this work.

| Code Sample | Chromophore | Add of Chromophore | Add of Mg(OH)$_2$ |
| --- | --- | --- | --- |
| LDH-CA | Carminic acid | During sol–gel synthesis | Non |
| LDH-HNB | Hydroxynaphthol blue | During sol–gel synthesis | Non |
| FLDH-CA | Carminic acid | During sol–gel synthesis | Non |
| FLDH-HNB | Hydroxynaphthol blue | During sol–gel synthesis | Non |
| FLDH-CA + MgO | Carminic acid | During sol–gel synthesis | Yes |
| FLDH-HNB + MgO | Hydroxynaphthol blue | During sol–gel synthesis | Yes |
| ME-FLDH-CA | Carminic acid | Memory effect | Non |
| ME-FLDH-HNB | Hydroxynaphthol blue | Memory effect | Non |
| ME-FLDH-CA + MgO | Carminic acid | Memory effect | Yes |
| ME-FLDH-HNB + MgO | Hydroxynaphthol blue | Memory effect | Yes |

*2.2. Characterization*

X-ray powder diffraction (XRD) patterns were obtained using a Bruker D8 advance diffractometer with a copper anode tube. The CuK$\alpha$ radiation ($\lambda$ = 1.54186 Å) was selected with a beam monochromator.

$^{27}$Al solid-state nuclear magnetic resonance (MAS NMR) spectra were obtained using a Bruker Avance II spectrometer at a Larmor frequency of 78.21 MHz. Spectra were acquired with short single pulses ($\pi/12$) and delay times of 0.5 s. The samples were spun at 10 kHz, and the chemical shifts were referenced to a 1M AlCl$_3$ solution.

The $^{19}$F MAS NMR spectra were measured by operating the spectrometer at 376.3 MHz using $\pi/2$ pulses of 6 $\mu$s with a recycle delay of 1 s; $^{19}$F chemical shifts were referenced to those of CFCl$_3$ at 0 ppm.

$^{13}$C CP MAS NMR spectra were collected at a frequency of 100.58 MHz using a Bruker Avance II HD 400 spectrometer. The spectra were acquired using a 4 mm cross-polarization (CP) MAS probe spinning at a rate of 5 kHz. We used typical $^{13}$C CP MAS NMR conditions for the $^{1}$H–$^{13}$C polarization experiment: a $\pi/2$ pulse of 4 $\mu$s, a contact time of 1 ms, and a delay time of 5 s. Chemical shifts were referenced to a chemical shift at 38.2 ppm of the CH signal of adamantane relative to TMS

*2.3. Formulation of the Pigmented Cream*

An emollient cream was prepared by the emulsification method [22,23], starting with stearic acid (14 g), almond oil (1 mL), NaOH (1 g), triethanolamine (1 mL), ethylenediamine tetra acetic acid (0.2 g), glycerin (6 mL), and methyl paraben (0.1 g). Briefly, the oil-soluble components and the emulsifier were melted in a water bath at 75 °C. Separately, water-soluble components were melted at 75 °C. After heating, the oil phase was moved into a mortar and pestle and, slowly, the water phase was added. Once the temperature had cooled down, the pigment was dispersed in the cream.

**3. Results**

*3.1. Pigments*

3.1.1. Effect of Fluorination in Colored LDH

Figure 1 displays XRD patterns of pigments; it can clearly be observed that the LDHs differ structurally depending on their chemical composition. The XRD pattern of the LDH-CA sample is one typical of an LDH synthesized by a sol–gel method; the peaks are broad and correspond to samples with a small size, which is expected for LDH samples synthesized by a sol–gel procedure. On the contrary, the analog fluorinated pigment, sample FLDH-CA, shows an XRD pattern with very low intense peaks matching to an LDH phase, suggesting that fluorine and carminic acid disordered the layered structure. An interesting result was found for the MEFLDH-CA sample, which has an XRD pattern that matches with a more ordered LDH; however, the peak at 38 degrees suggests that a periclase mixed oxide phase is present; in this context, it should be remembered that this pigment was prepared by the treatment of fluorinated LDH and rehydration in carminic acid. Thus,

it seems that the presence of carminic acid in rehydration inhibits the rehydration. Lastly, the XRD patterns of the samples synthesized with Mg(OH)$_2$ addition exhibited peaks distinctive of the MgO phase at 38.1 and 62 degrees.

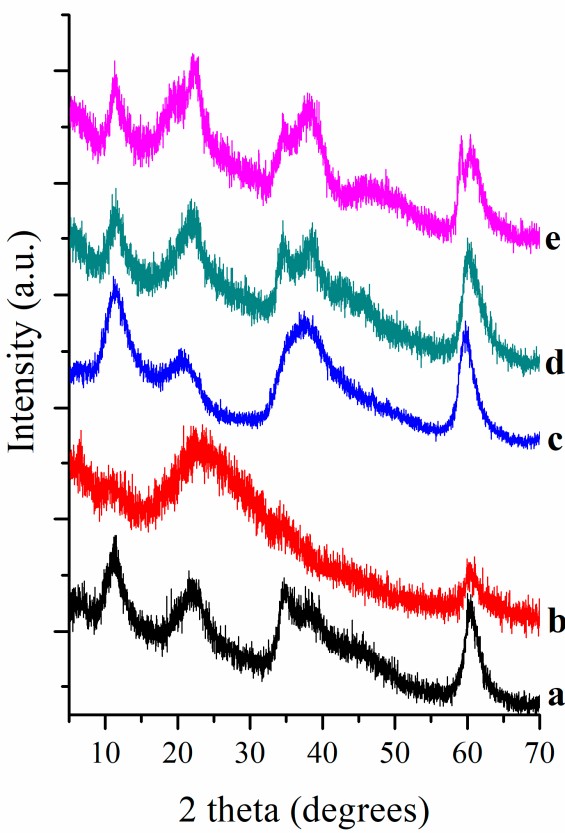

**Figure 1.** XRD patterns of pigments generated by sorption of carminic acid onto layered double hydroxides. (**a**) LDH-CA, (**b**) FLDH-CA, (**c**) FLDH-CA + MgO, (**d**) ME-FLDH-CA, and (**e**) ME-FLDH-CA + MgO.

Figure 2a shows that the $^{13}$C CP MAS NMR spectrum of CA, as a part of hybrid pigments, changed significantly compared with the spectrum of pure CA. The peaks due to the resonances of aromatic carbons (168–188 ppm) in the pigments are significantly broader and less intense than in pure CA. In the 62–87 ppm range, the signals due to the glucose unit [24] are also a little broader in pigments. These results suggest that the CA molecule is partially immobilized in the pigment, in other words, an interaction between the aromatic part of CA and the layers of the LDH seemingly takes place. The $^{27}$Al MAS NMR (Figure 2b) spectra of all colored and white samples are composed of a single peak centered around 9 ppm, which is due to the six-fold coordinated aluminum [25], in line with a pure hydrotalcite structure. No significant differences were observed in spectra as a function of the compositions of the pigments.

The $^{19}$F NMR spectra included in Figure 2c confirm that fluorine does not change its chemical environment when the white sample (FLDH) becomes the colored one (FLDH-CA). For both samples, only one single peak was observed at −168.1 ppm, assigned to AlF$_{6-x}$O$_x$, with x taking values from 1 to 6 [26]. In contrast, when the colored sample was coated with magnesium hydroxide, the resonance peak shifted to high fields, specifically to −163.8 ppm, supporting that fluorine is redistributed, leading to fluorinated and oxygenated magnesium species, which is an expected result, as the number of oxygen atoms is higher in FLDH-CA than in FLDH-CA + MgO.

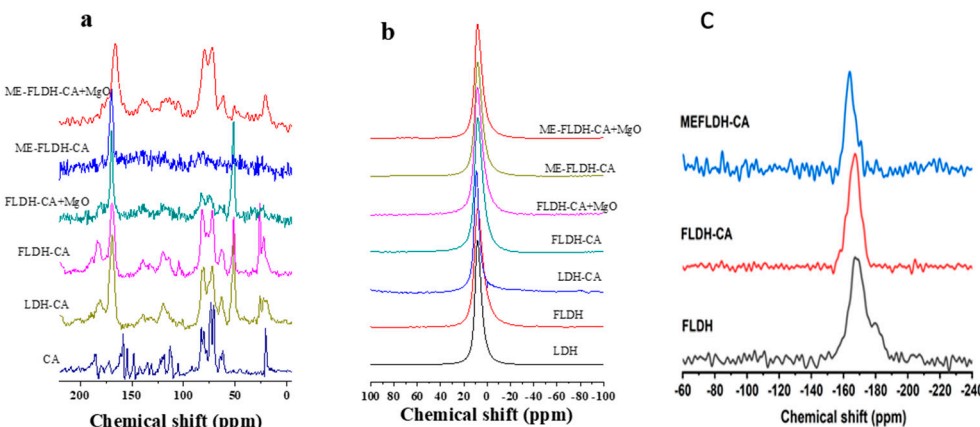

**Figure 2.** Multinuclear solid-state NMR spectra of pigments. (**a**) $^1$H→$^{13}$C CP/MAS NMR, (**b**) $^{27}$Al MAS NMR, and (**c**) $^{19}$F MAS NMR.

3.1.2. Loading Chromophores during Sol–gel Synthesis versus Memory Effect

As described in Section 2, the pigments were obtained by incorporating CA or HNB into the LDHs. Two ways for this loading were tested: on the one hand, the chromophore was loaded during sol–gel synthesis, and on the other hand, each LDH was thermally treated and rehydrated, and the chromophore was added during the rehydration step. Figure 3 displays pictures of the fresh pigments obtained. Note that the method used to load chromophore influences the final color of the pigments. The pigments FLDH-CA and ME-FLDH-CA differ significantly in color, as visible by the human eye. Loading chromophores during sol–gel synthesis leads to a purple pigment, and loading chromophores in the rehydration step during the memory effect of LDH results in a red pigment. After coating with magnesium hydroxide, both pigments had a similar purple color. When the chromophore was HNB, the color also differed between the FLDH-HNB and ME-FLDH-HNB pigments and became similar after recovery with the magnesium phase.

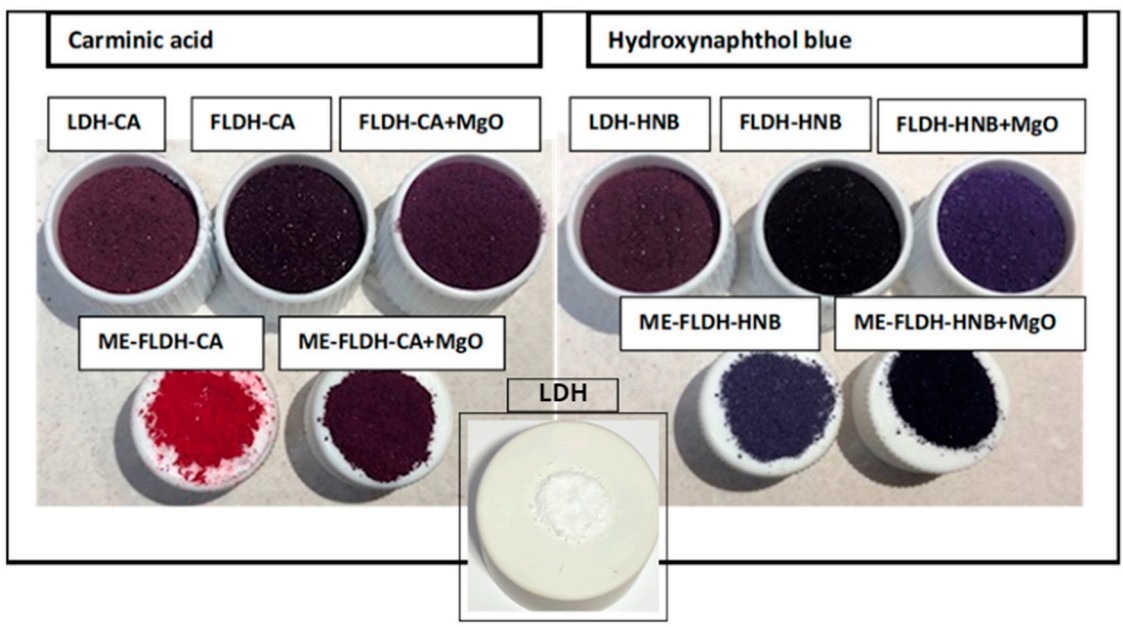

**Figure 3.** Pigments generated by the sorption of carminic acid onto layered double hydroxides. As a reference, a white LDH powder is included.

The UV-vis spectra of pigments containing carminic acid are included in Figure 4. A broad band centered at 450 nm composes the spectrum of reference, that of pure carminic acid. For pigments LDH-CA and FLDH-CA, two bands are observed at 340 and 555 nm. These two bands are often observed when carminic acid is treated with mordents based on aluminum. The presence of fluorine in LDH caused the UV-vis band to slightly shift to red. It was earlier [21,27] pointed out that a consequence of the fluorination of a magnesium–aluminum hydrotalcite is the formation of $MgO_{6-x}F_x$ species, which, in turn, induces the creation of strong basic sites, as well as aluminum with low coordination numbers. These sites should act as the adsorption sites for CA. However, this interaction is not strong enough, as the absorption wavelength is easily modified by the presence of $Mg(OH)_2$. The presence of $Mg(OH)_2$ in these two pigments leads the color of the band to shift to red. More interestingly, the spectra of the pigments prepared by the memory effect also show two broad absorption bands centered at 355 and 562 nm. They shift to red compared with the bands observed in the pigments prepared with chromophore loading during sol–gel synthesis.

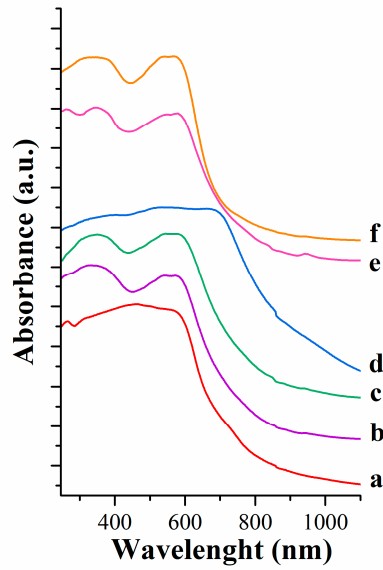

**Figure 4.** UV-vis spectra of carminic acid and pigments generated by sorption of carminic acid onto layered double hydroxides. (**a**) CA, (**b**) LDH-CA, (**c**) FLDH-CA, (**d**) FLDH-CA + MgO, (**e**) ME-FLDH-CA, and (**f**) ME-FLDH-CA + MgO.

### 3.1.3. Aging of Pigments

Fresh pigments were stored at room temperature for ten years and then characterized again in order to verify any changes regarding structure and color. The XRD patterns of the aged pigments were close to that of the fresh pigments, as Figure 5 shows. Only one pigment evolved within the XRD limit of detection: the XRD of the aged pigments seems to be closer to the hydrotalcite structure in contrast to the fresh one, which was close to the MgO periclase structure. The $^{13}C$ CP MAS NMR spectra of the aged pigments are displayed in Figure 6. When comparing the spectra of LDH-CA and FLDH-CA + MgO (Figure 2a) with their respective aged samples, no important changes were observed; only the signal that appears near 170 ppm shifted 3 ppm to a higher field with aging, indicating that group C=O fades as time goes on, in other words, the interaction of CA with LDH changes slightly.

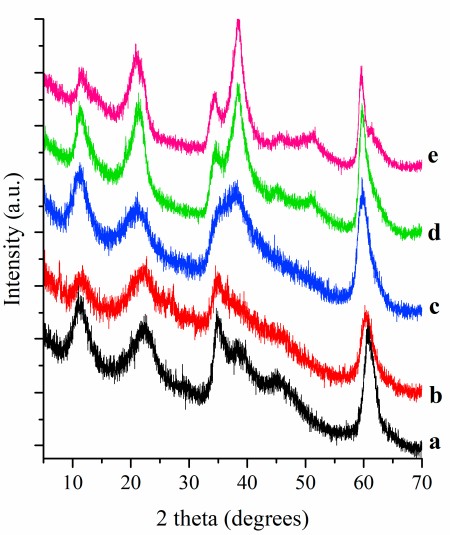

**Figure 5.** XRD patterns of pigments aged for 10 years. (**a**) LDH-CA, (**b**) FLDH-CA, (**c**) FLDH-CA + MgO, (**d**) ME-FLDH-CA, and (**e**) ME-FLDH-CA + MgO.

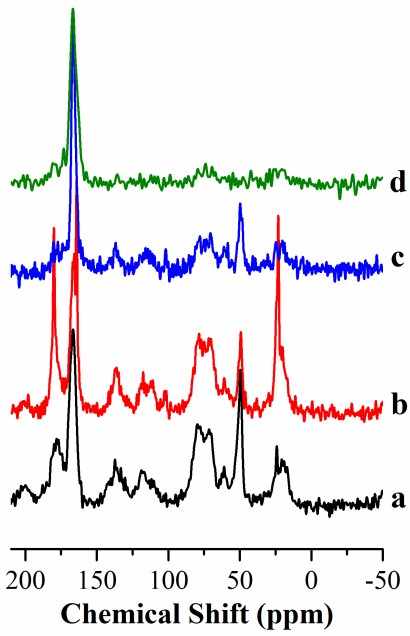

**Figure 6.** $^1$H→$^{13}$C CP/MAS spectra of aged pigments. (**a**) LDH-CA, (**b**) FLDH-CA, (**c**) FLDH-CA + MgO, and (**d**) ME-FLDH-CA + MgO.

The spectrum of the FLDH-CA sample also changed once it was aged. In the interval of 62–87 ppm, where the glucose unit signals appear, a decrease in intensity is observed, and the peaks are no longer well defined once the sample ages. This is probably because the glucose unit interacts with LDA via the –OH groups, resulting in the glucose unit having some rigidity.

In general, the sample that changed the least as a result of aging was the sample with the Mg(OH)$_2$ coating, while the FLDH-AC sample presented the most changes. Therefore, the coated hybrid pigment is the most stable under moderate changes in temperature, humidity, and radiation.

More significant differences in color between the aged and fresh pigments were observed. Color was measured according to the CIEL*a*b* parameters [28]. Color fading was calculated as follows:

$$\Delta E_{ab}^* = \sqrt{\left(L_2^* - L_1^*\right)^2 + \left(a_2^* - a_1^*\right)^2 + \left(b_2^* - b_1^*\right)^2}$$

where L is the brightness parameter, and a and b the color coordinates.

### 3.2. Pigmented Cream

Figure 7 includes photographs of emollient cream pigmented with several of the pigments prepared in this work. The image was updated as time passed.

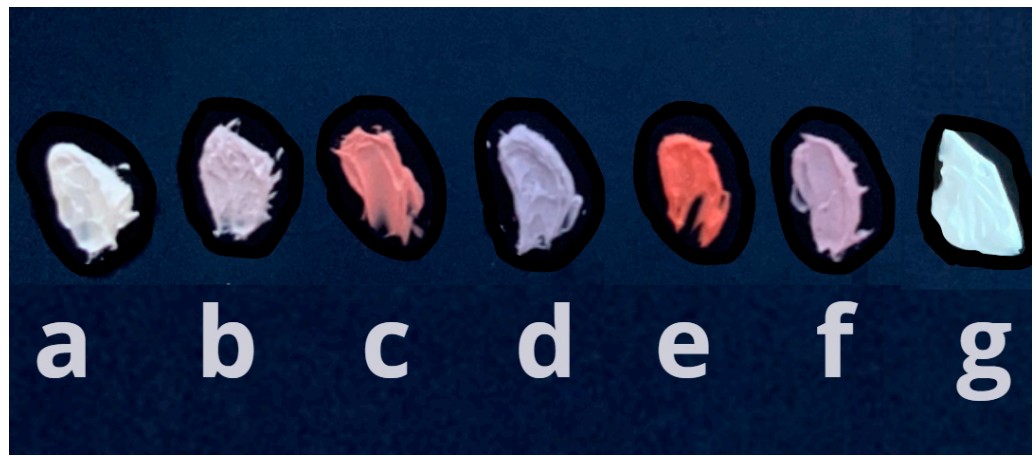

**Figure 7.** Pigmented emollient creams prepared from white cream (**a**) and several pigments. (**b**) LDH-CA, (**c**) FLDH-CA, (**d**) FLDH-CA + MgO, (**e**) ME-FLDH-CA, and (**f**) ME-FLDH-CA + MgO. As a reference, the white pigment made using FLDH is shown in (**g**).

Fresh colored creams prepared with hybrid pigments maintained the same consistency as the white cream. The color acquired by the creams was, at beginning, the same as that of the pigments used in coloring. However, as time passes, the LDH-CA pigmented cream fades slightly, and small red aqueous droplets are secreted (aqueous phase confirmed by FTIR spectroscopy).

## 4. Discussion

The presence of carminic acid in pigments does not collapse the layered structure of hydrotalcite, which acts as a host of chromophores, and the main interaction between CA and LDH is facilitated through the aromatic rings of CA and the hydroxylated layers of LDH. When fluorine is present in LDH, it can play an important stabilizing role in the magnesium phase coating, as discussed below.

The results in pigments containing hydroxy naphthol blue were similar to those found for the carminic acid series; the interaction with the aromatic part of HNB was also confirmed, and the role of fluorine in the magnesium phase in the coated pigments was also observed. For the sake of brevity, the corresponding NMR spectra and XRD patterns of the HNB series are presented in the Supplementary Information (Figures S1 and S2).

Regarding the textural properties, all the pigments were non-porous, with a specific surface area (SSA) between 1 and 7 m$^2$/g (Table S1, Supplementary Information). Considering that the white samples of LDH and FLDH have SSAs of up to 310 and 318 m$^2$/g, respectively, it should be concluded that CA and HNB molecules are sufficient for blocking the pores of LDH.

The $^{27}$Al NMR MAS and UV-Vis results suggest a stronger CA–aluminum interaction in the pigments prepared by the memory effect than that observed in pigments LDH-CA

and FLDH-CA. It was earlier [21,27] pointed out that a consequence of the fluorination of a magnesium–aluminum hydrotalcite is the formation of $MgO_{6-x}F_x$ species, which, in turn, induces the creation of strong basic sites, as well as aluminum with low coordination numbers. These sites should act as the adsorption sites for CA. However, this interaction is not strong enough, as the absorption wavelength is easily modified by the presence of $Mg(OH)_2$. The presence of $Mg(OH)_2$ in these two pigments leads to a shift in the color of the band to red; it seems that the presence of $Mg(OH)_2$ increases the basicity of the material, leading to purple pigments. In the absence of $Mg(OH)_2$, the pigments keep the color as synthesized, and the color persisted after they were dispersed in emollient creams, as shown in Figure 7. More interestingly, the spectra of the pigments prepared by the memory effect also show two broad absorption bands centered at 355 and 562 nm. They shift to red compared with the bands observed in the pigments prepared with chromophore loading during sol–gel synthesis. The colors of pigments FLDHCA and ME FLDHCA differ greatly. It should be emphasized that the memory effect includes the thermal treatment of LDH, thus creating centers of unsaturated aluminum sites, such as four-fold coordinated $Al^{3+}$ [29,30]. Thus, this species can act as CA adsorption sites since CA have several dipolar functional groups (structure of CA in Figure S3).

The $\Delta E_{ab}^*$ is reported in Table 2. The higher the $\Delta E_{ab}^*$, the greater the loss of color. Overall, the pigments were stable; the $\Delta E_{ab}^*$ values were < 20. Small changes in color were observed: For example, the sample with a major loss of color was LDH-CA, but the fluorinated pigment significantly diminished color fading; the presence of the recovery magnesium phase also inhibits the loss of color. All pigments containing the magnesium phase coating showed a decrease in the $\Delta E_{ab}^*$ as compared to the uncoated pigment. Interestingly, the pigments synthesized using the memory effect were the pigments where the recovery with $Mg(OH)_2$ was less significant. Actually, the color faded similarly with or without $Mg(OH)_2$, in line with the notion that pigments prepared by the memory effect contain an immobilized chromophore in coordinated unsaturated aluminum sites. Note, however, that the pigments are discolored significantly, as hydration leads to a layered hydrotalcite-like structure, as shown by the XRD results.

**Table 2.** $\Delta E_{ab}^*$ as calculated from parameters of brightness and color, applying the formula expressed on page 8. Fresh and aged in the table correspond to state 1 and 2 in the formula, respectively.

| Sample | L | | a | | b | | $\Delta E_{ab}^*$ |
|---|---|---|---|---|---|---|---|
| | Fresh | Aged | Fresh | Aged | Fresh | Aged | |
| LDH-CA | 20 | 24.3 | 47 | 61.3 | 8 | 15.1 | 16.53 |
| FLDH-CA | 19.9 | 21.2 | 21 | 20 | 8.3 | 9.8 | 2.34 |
| FLDH-CA + MgO | 21.5 | 23.6 | 40.8 | 51 | 7.1 | 13.2 | 12.06 |
| LDH-HNB | 19.3 | 22.2 | 20.5 | 27.5 | 8.5 | 10 | 7.72 |
| FLDH-HNB | 19.2 | 22 | 20.4 | 23.5 | 8.3 | 8.9 | 4.22 |
| FLDH-HNB + MgO | 24.2 | 23.4 | 36.8 | 34 | 14.4 | 14 | 2.93 |
| ME-FLDH-CA | 20.7 | 27.6 | 51.3 | 62.4 | 5.8 | 15.2 | 16.17 |
| ME-FLDH-CA + MgO | 23.2 | 29.2 | 34 | 48 | 9.9 | 6.6 | 15.58 |
| ME-FLDH-HNB | 22.3 | 25 | 28.3 | 35 | 6.7 | 11.1 | 8.45 |
| ME-FLDH-HNB + MgO | 22.3 | 26 | 29.9 | 33.8 | 8.9 | 11.8 | 6.11 |

Changes observed in the cream pigmented with LDH-CA can be interpreted as the release of the chromophore from the pigment and dissolution in the water contained in the cream. The cream recovered its original appearance when it was mixed for 15 s. All other creams were unchanged for periods as long as 4 years. This result is in line with the most stable pigments that were prepared with fluorinated LDH and improved with the $Mg(OH)_2$ phase.

## 5. Conclusions

Carminic acid and hydroxynaphthol dyes were adsorbed on a hydrotalcite support through weak interactions, leading to the emergence of hybrid pigments. However, dye is slightly removed from hydrotalcite when it comes into contact with water, but when $Mg(OH)_2$ is incorporated as a hybrid pigment coating, it interacts with the surface of the dye, functioning as a kind of shell to prevent the dye chromophore from release.

The adsorption of the dyes drastically decreases the surface area of the hydrotalcite, evidencing that the dye occupies most of the available specific area; however, the presence of fluorine or magnesium hydroxide does not help to develop large surface areas.

The hybrid pigments showed slight structural and textural changes; however, the changes were not drastic.

The pigments obtained by the memory effect were the ones that underwent the most changes in their surface properties and porosity.

The results obtained in colorimetry reveal that the hybrid pigments obtained using sol–gel do not show a great difference in color before and after being subjected to aging, especially for coated samples. However, the pigments obtained by the memory effect are the most discolored. This means that the pigments obtained using sol–gel remain stable under an air atmosphere for periods as long as ten years, since their color remains almost unaltered.

**Supplementary Materials:** The following supporting information can be downloaded at: https://www.mdpi.com/article/10.3390/colorants3020009/s1. Figure S1: XRD patterns of pigments prepared from combination of layered double hydroxides and hydroxinaphtol blue. Figure S2: $^{13}$CP/MAS NMR and $^{19}$F MAS NMR spectra of pigments prepared from combination of layered double hydroxides and hydroxinaphtol blue; Figure S3. Chemical structure of hydroxinaphtol blue (left) and carminic acid (right); Table S1: Textural properties of white layered double hydroxides and their respective pigments.

**Author Contributions:** Formal analysis, M.H.; methodology, C.F.; writing—original draft preparation, M.H. and A.G.-V.; supervision, J.L.R.; conceptualization, supervision, and project administration, E.L. All authors have read and agreed to the published version of the manuscript.

**Funding:** This work was financially supported by CONACYT Mexico (Grant 220436).

**Institutional Review Board Statement:** Not applicable.

**Informed Consent Statement:** Not applicable.

**Data Availability Statement:** The original contributions presented in the study are included in the article/Supplementary Materials, further inquiries can be directed to the corresponding authors.

**Conflicts of Interest:** The authors declare no conflict of interest.

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
