# Peer review of "Highly Stable Hybrid Pigments Prepared from Organic Chromophores and Fluorinated Hydrotalcites"

_2079-6447, doi:10.3390/colorants3020009_

Round 1
Reviewer 1 Report
Comments and Suggestions for Authors
This paper concerns hybrid color materials combining layered double hydroxides and organic dyes.
It is very interesting and deserves to be published, but I have a few questions.
1. The colors differ greatly between FLDH-CA and ME-FLDH-CA, please explain why such a large difference occurs with and without the layered structure.
2. If the layered structure produces vivid colors due to the memory effect, but is unstable and is not stabilized by the addition of MgO, is this hybrid colorant not suitable for practical use?
3. In Figures 1, 4, 5, 6, and 7, aren't ME-LDH-CA and ME-LDH-CA+MgO correctly ME-FLDH-CA and ME-FLDH-CA+MgO, respectively?
Author Response
Reviewer 1
This paper concerns hybrid color materials combining layered double hydroxides and organic dyes.
It is very interesting and deserves to be published, but I have a few questions.
- The colors differ greatly between FLDH-CA and ME-FLDH-CA, please explain why such a large difference occurs with and without the layered structure.
Authors: Thanks for the observation, The explanation is related to the presence of acid sites in sample ME-FLDH CA. An explanation was included in lines 277-284.
- If the layered structure produces vivid colors due to the memory effect, but is unstable and is not stabilized by the addition of MgO, is this hybrid colorant not suitable for practical use?
Authors: Yes, the hybrid pigment is suitable for practical use as the pigment is dispersed in a cream and the cream maintains color for longtime. A statement in lines 272-277 was included.
- In Figures 1, 4, 5, 6, and 7, aren't ME-LDH-CA and ME-LDH-CA+MgO correctly ME-FLDH-CA and ME-FLDH-CA+MgO, respectively?
Authors: We are sorry for this mistake. The text refers correctly to ME-FLDH-CA and ME-FLDH-CA+MgO. The legend to figures was corrected. Thank you for observation.

Reviewer 2 Report
Comments and Suggestions for Authors
Dear Authors,
The manuscript describes the preparation of hybrid pigments involving lamellar double hydroxide and dyes. I consider it necessary to clarify some points and improve the text before it is accepted.
i) Abstract (very short): it is necessary to include information about the prepared pigments and a conclusion about application.
ii) Why is there no data on white HDL in the characterization of pigments? XRD data, UVVis, etc.
iii) In the comparative images (Figures 3 and 7) you could place a photo of white HDL next to it, after all it is the basis of hybrid pigments.
iv) What are the criteria for choosing the dyes carminic acid and hydroxynaphthol blue? What is the main application of these dyes? It could include their structural formulas, molar mass, etc.
v) Vibrational spectroscopy (FTIR) would help show what happens to dyes and HDL.
vi) Table 2 presents DeltaE data, how was this calculation made? I suggest you present a complete table with the data used to generate DeltaE. I suggest consulting examples of manuscripts that discuss the preparation of hybrid pigments involving a lamellar solid and dye

Dear Authors,
I suggest a technical/grammatical review of the text.
Author Response
Reviewer 2
The manuscript describes the preparation of hybrid pigments involving lamellar double hydroxide and dyes. I consider it necessary to clarify some points and improve the text before it is accepted.
- i) Abstract (very short): it is necessary to include information about the prepared pigments and a conclusion about application.
Authors: Abstract was modified in line with comment of reviewer.
- ii) Why is there no data on white HDL in the characterization of pigments? XRD data, UVVis, etc.
Authors: characterization of HDL and FHDL is well-known, actually many journals consider routine work. We have included the reference (21) where a full characterization was reported.
iii) In the comparative images (Figures 3 and 7) you could place a photo of white HDL next to it, after all it is the basis of hybrid pigments.
Authors: Thank you for observation, we have modified figures 3 and 7 in response to the suggestion.
- iv) What are the criteria for choosing the dyes carminic acid and hydroxynaphthol blue? What is the main application of these dyes? It could include their structural formulas, molar mass, etc.
Authors: Carminic acid and hydroxynaphtol blue are two dyes that are greatly used in food and cosmetic industry. A short text was included in lines 52-57. The structure of both dyes is included as supporting information (Figure S3).
- v) Vibrational spectroscopy (FTIR) would help show what happens to dyes and HDL.
Authors: We have recorded FTIR spectra, unfortunately they do not provide additional information that obtained by NMR and included in manuscript.
- vi) Table 2 presents DeltaE data, how was this calculation made? I suggest you present a complete table with the data used to generate DeltaE.
Authors: Table 2 was modified as suggested by reviewer. In revised version the L*a*b parameters were included. The caption to table indicates that the formula in page 7 was applied to calculate DeltaE where state “2” and “1” correspond to aged and fresh sample respectively.

Round 2
Reviewer 1 Report
Comments and Suggestions for Authors
The authors revised the manuscript appropriately according to the peer review comments. I think the manuscript is acceptable.
Reviewer 2 Report
Comments and Suggestions for Authors
Dear Autors,
Congratulations on the manuscript. A good contribution to Colorants.